# An Improved Droop Control Strategy for Grid-Connected Inverter Applied in Grid Voltage Inter-Harmonics and Fundamental Frequency Fluctuation

**Wanwan Xu** [1], **Bin Wang** [1], **Jiang Liu** [1,2,*] **and Da Li** [3]

1   School of Information Science and Engineering, Wuhan University of Science and Technology, Wuhan 430081, China; rosywan0114@163.com (W.X.); binwang907@163.com (B.W.)
2   Engineering Research Center for Metallurgical Automation and Measurement Technology of Ministry of Education, Wuhan University of Science and Technology, Wuhan 430081, China
3   Shenzhen Hopewind Electric Co., Ltd., Shenzhen 518127, China; lidamoan@163.com
*   Correspondence: liujiang@wust.edu.cn; Tel.: +86-027-6886-2520

**Abstract:** This paper presents a current suppression method based on a droop control strategy under distorted grid voltage with inter-harmonics and fundamental frequency fluctuation. In this proposed strategy, the current incomplete derivation controller is employed to decrease the negative impact caused by harmonic and inter-harmonic grid voltage. This method provides a good dynamic response and has low complexity against the inter-harmonics with unfixed fundamental frequency. Based on a mathematical model of the grid-connected inverter, we designed novel instantaneous frequency detection and feed-forward methods to suppress the grid fundamental frequency fluctuation impacts. Then the main parameters were analyzed. The simulation and experimental results verified the feasibility and effectiveness of the proposed method.

**Keywords:** droop control; grid-connected; inter-harmonics; incomplete derivation; instantaneous frequency detection

## 1. Introduction

Droop control has been widely used for microgrid inverters, but its performance is rarely considered for future electronic-based power systems. There is an increasing number of micro-source electronic power devices being integrated into the grid. Abundant wide-frequency signals, such as inter-harmonics (not an integer multiple of the fundamental frequency), and fundamental fluctuations are injected into power grid [1]. In addition, wind turbines and photovoltaic units are often installed far from load centers (e.g., offshore wind farms). These weak interfaces have high impedance, which affects power stability and voltage regulation [2].

The droop control strategy has been widely investigated under ideal grid voltage conditions, [3]. However, harmonic voltage is becoming more common in the power grid. Distorted voltage conditions, which include frequency and amplitude fluctuates and harmonics, are worst at the point of common coupling (PCC) [4]. The power flow of droop control is sensitive to the grid frequency [1]; if it is not modified under distorted grid voltage conditions, the grid current becomes distorted, leading to power oscillations [5]. In order to satisfy the requirements of total harmonic distortions (THD), the harmonics current should be suppressed.

Several strategies have been proposed to suppress harmonic current under integer harmonic voltage. In Reference [6], the current loop gain at harmonic frequency was increased, by a proportional resonant plus multiple resonant harmonic compensators, to achieve infinite loop gains at the target harmonic. This is an appropriate strategy for harmonics but ignored inter-harmonics. In Reference [7], the repetitive control was proposed to suppress 6n±1 order harmonics currents under the harmonic grid. However,

repetitive control has a slow response as it is a fractional denominator. In Reference [8], a current-detection-based harmonic compensation method was proposed with active power filtering capabilities. But this is a difficult and expensive method when applied to a real power grid to load current harmonic detection. Recently, a model predictive controller was used to improve operation of inverters [9], however, this approach has high computational complexity and requires accurate parameters for the inverter system. In Reference [10], the paper analyzed harmonic power sharing caused by mismatched grid impedance for island micro-grid, then developed a two-dimensional impedance shaping control. Considering circulation current or instability problem in multiple inverters, we only considered the harmonic suppression of a single interlinking inverter.

In addition to harmonic components, there are inter-harmonic voltage components in a real power grid. Inter-harmonic voltage can be caused by variable-frequency loads, photo-voltaic converters [11], adjustable speed device, etc. The International Electronical Commission recommended considering inter-harmonics as part of the assessment criteria for inverter systems. However, few studies have focused on the influence of inter-harmonic voltage on the inverter system.

The existing harmonic current suppression based on repetitive control or a resonant regulator works on integer harmonic voltage effectively, but this is not a valid approach for inter-harmonic voltage. Meanwhile, this method considers the fundamental frequency as fixed. Some harmonic extracting methods, such as Kalman filtering, are still difficult to extend to inter-harmonic extraction [12]. In Reference [13], a stator harmonic current suppression method was proposed for a DFIG system considering inter-harmonics, however, droop control is not suitable for a DFIG system. In Reference [14], a selective inter-harmonic filter was introduced to a classical control scheme, which made it possible to compensate inter-harmonics in a proper band around fundamental frequency.

Although the above techniques were employed for harmonic suppression, they only effectively work on integer harmonic voltage with fixed fundamental frequency, which will be invalid on inter-harmonic voltage. In order to enhance the capability for suppression of inter-harmonic current for a grid-connected inverter with droop control strategy, this paper presents a harmonic current suppression strategy for a grid-connected inverter based on incomplete current derivation feedback. It has been validated within a harmonic frequency range instead of at frequency points. Compared with the traditional droop control strategy, the value of frequency feed-forward is calculated by the rotating angular velocity of the voltage vector instead of the phase-locked loop (PLL) with fundamental frequency variations. The current feedback is obtained by an incomplete derivation with a high-pass filter for suppression of the harmonic and inter-harmonic current caused by the distorted grid. Moreover, unlike the traditional droop control strategy without current and instantaneous frequency feedback, the proposed strategy has better performance in particular fundamental frequency variations.

The remainder of this paper is organized as follows. In Section 2, the structure and droop control of the grid-connected inverter is described, and the influence of grid voltage harmonics and inter-harmonics on frequency detection are analyzed. In Section 3, the improved suppression control strategy is explained in detail. In Section 4, simulation based on Matlab/Simulink and the experimental lab results from a 10 kW three-phase micro-grid inverter are provided to verify the effectiveness of the proposed improved harmonic current suppression control strategy. Finally, Section 5 concludes this paper.

## 2. Modeling of Grid-Connected Inverters

### 2.1. System Modeling

The basic topology of an AC micro-grid with three phase inverter is shown in Figure 1. This inverter architecture is a conventional full-bridge inverter topology, which is suitable for grid-connected and stand-alone operation in a micro-grid. Several PV modules are connected to the DC side, which is interfaced with the grid through an LCL filter and mode switch. The inverter is controlled by droop control strategy through the space vector

pulse width modulator. The main load laminator heating system and vacuum mixer have characteristics of high power, which lead to amplitude and frequency fluctuates of the grid side voltage. The use of three-phase induction heaters could lead to power drops and an imbalance of grid voltage. Additional equipment to compensate for the harmonics, reactive power, or voltage imbalance would add a complex interaction to the electronic-based power system [15,16].

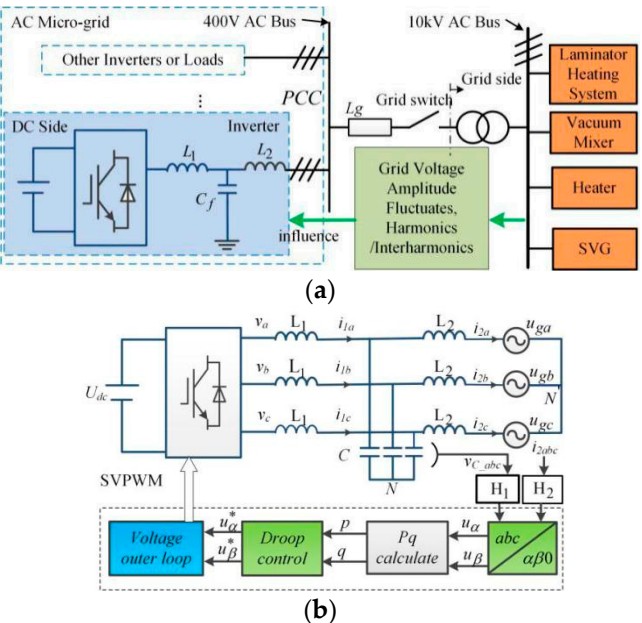

**Figure 1.** (**a**) Topology of the AC micro-grid in a photo-voltaic panel factory; (**b**) control system structure of the inverter in a grid-connected mode (List of symbols is in Appendix A).

For the control system structure shown in Figure 1, there is no zero-sequence current in the three-phase three-wire grid-connected inverter. Hence, the system is usually controlled in the static coordinate system.

The droop control method comes from an attempt to mimic the self-regulation capability of a synchronous generator. It is used to adjust frequency and amplitude of $V_{ref}$ according to distribution of $P$ and $Q$. It is assumed that the power angle has a small value. The active and reactive powers delivered to the grid can be written as [17]:

$$P = \frac{V_{abc}}{R_g^2 + X_g^2}[R_g(V_{abc} - V_g \cos\delta) + X_g V_g \sin\delta] \underset{\substack{R\approx 0 \\ \delta\to 0}}{\approx} \frac{V_{abc} V_g}{X_g}\sin\delta \tag{1}$$

$$Q = \frac{V_{abc}}{R_g^2 + X_g^2}[X_g(V_{abc} - V_g \cos\delta) - R_g V_g \sin\delta] \underset{\substack{R\approx 0 \\ \delta\to 0}}{\approx} \frac{V_{abc}(V_{abc} - V_g \cos\delta)}{X_g} \tag{2}$$

When inductance is much greater than resistance under a weak grid, voltage amplitude depends on reactive power, while frequency depends on active power, then the droop control expression can be written as (3) and (4).

$$f - f_0 = -k_p(P - P_{ref}) \tag{3}$$

$$V - V_0 = k_q(Q - Q_{ref}) \tag{4}$$

As shown in Figure 1b, the instantaneous active power and reactive power are calculated from the measured-out voltage and current throughput. Using Clark transformation,

$\alpha\beta$ coordinate, and instantaneous power value, the power is passed through a low-pass filter with cutoff frequency of $\omega_c$, and $P$ and $Q$ can be obtained as (5).

$$
\begin{bmatrix} P \\ Q \end{bmatrix} = \frac{\omega_f}{s+\omega_f} \begin{bmatrix} u_d & u_q \\ u_q & -u_d \end{bmatrix} \begin{bmatrix} i_d \\ i_q \end{bmatrix}
$$
$$
\omega = \omega_0 + k_p \omega_p (P_0 - P)
$$
$$
u_d = E_0 + k_q (Q_0 - Q), u_q = 0
$$
(5)

where $k_p$ and $k_q$ are the droop coefficient; $P_0$ and $Q_0$ are reference values of active power and reactive power; and $\omega_0$ and $E_0$ are the fundamental frequency and voltage.

In a previous study [1], the small-signal linearizing mode of the droop-controlled inverter was manipulated and the results showed that the active power is significantly greater and less sensitive to fluctuation of grid voltage magnitude. When the droop coefficient $k_p$ is fixed, the active power has a constant sensitivity to grid frequency fluctuation. Therefore, grid fluctuation has a significant impact on power flow control in the grid-connected mode.

### 2.2. Instantaneous Frequency Detection

DG units are typically a long distance from the grid. Consequently, grid voltage magnitude and frequency are replaced by filter capacitor voltage through PLL. PLL is a process of extracting information about the frequency and phase angle of the fundamental frequency positive sequence component of grid voltage [18]. The performance of the conventional synchronous reference frame phase-locked loop (SRF-PLL) has been widely used [19], Figure 2 shows a system of a three-phase SRF-PLL. The system consists of a phase detector, a loop filter, and a digital-controlled oscillator. In the presence of non-sinusoidal and unbalanced conditions, problems related to calculating the root mean square and power measurement have generated research interest in the literature [20]. The phase angle extraction of PLL is affected by various grid voltage disturbances, including imbalances, harmonic distortion, direct current, frequency variations, and inter-harmonics [21].

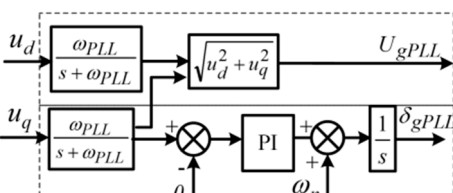

**Figure 2.** System of three-phase SRF-PLL.

In Figure 2, the proportional integral (PI) controller is used to ensure that the grid voltage vector is perfectly aligned along the $q$-axis. Then, this is added with the nominal frequency of the grid and fed to the integrator, which acts on the frequency error signal and output phase angle value. This phase angle is fed back to the transformation block to complete a closed control loop [22].

Lower bandwidth of SRF-PLL can improve phase estimation accuracy, however it will result in a slow dynamic response under distorted grid conditions [23]. When harmonics fluctuate rapidly, real-time performance of frequency detection will be unsteady. A Bode diagram of SRF-PLL is shown in Figure 3, it shows the phase and amplitude decreases in the high-frequency range. Therefore, high-frequency fluctuations cannot be measured by SRF-PLL accurately. An enhanced proportion coefficient can improve the range of detection frequency, however, the gain margin would decrease as well, which would lead to oscillation [24,25].

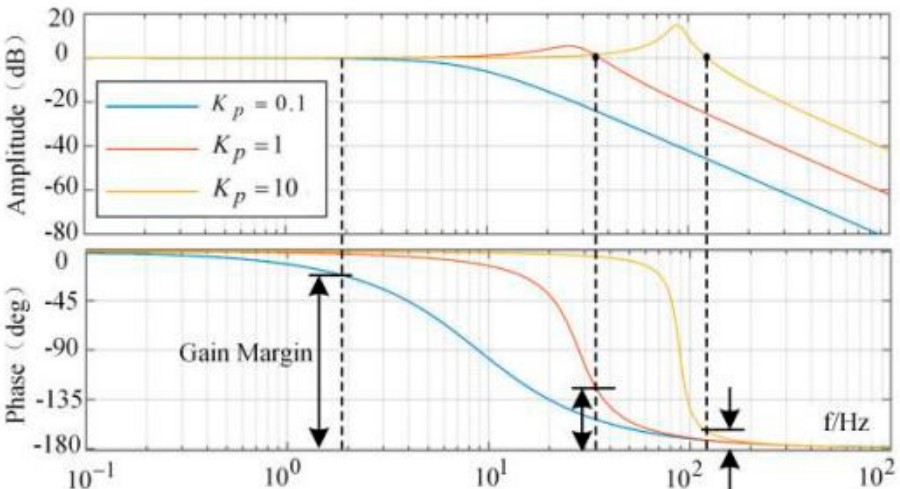

**Figure 3.** Bode diagram of SRF-PLL with different $k_p$.

In this paper, we propose a new method for accurate and fast frequency measurement. The instantaneous frequency can be calculated through the rotating angular velocity of the voltage vector. Figure 4 shows a rotation vector diagram of grid side voltage at two adjacent sampling times measured by Clark transformation.

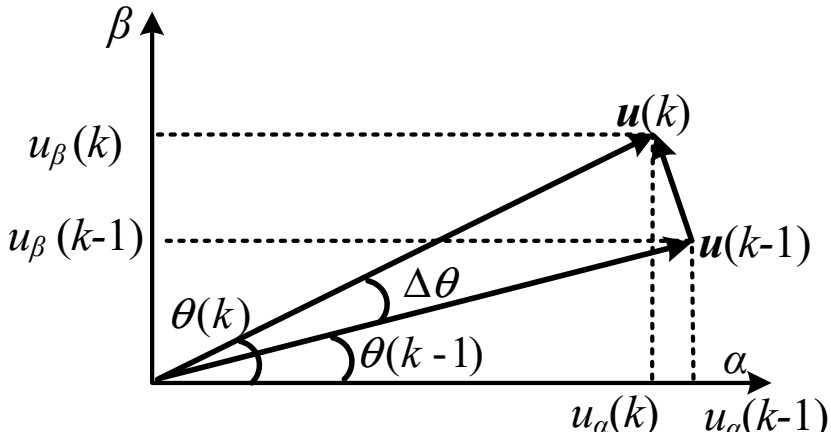

**Figure 4.** Rotation vector diagram of grid side voltage at adjacent sampling times.

Where, $u(k)$ and $u(k-1)$ are voltage vectors at two adjacent sampling times, $\theta(k)$ and $\theta(k-1)$ are voltage vector angles. Where, $\Delta t$ is sampling time and m is the maximum harmonic order of the grid, instantaneous frequency of voltage can be calculated as:

$$f = \frac{\Delta\theta}{2\pi\Delta t} \approx \frac{\sin\Delta\theta|_{\lim\Delta\theta\to 0}}{2\pi\Delta t} = \frac{\sin(\theta(k)-\theta(k-1))}{2\pi\Delta t} = \frac{\sin\theta(k)\cos(k-1)-\cos(k)\sin(k-1)}{2\pi\Delta t} \tag{6}$$

$$\sin\theta(k) = \frac{u_\beta(k)}{|u(k)|} \tag{7}$$

$$\cos\theta(k) = \frac{u_\alpha(k)}{|u(k)|} \tag{8}$$

$$\Delta t \leq \frac{1}{2} \times \frac{1}{m \times 50} \tag{9}$$

## 3. Control Strategy Based on Incomplete Current Differential Feedback

VVF techniques have been developed by improving throughput dynamics and self-stability [26] and system electrical damping [27]. The grid voltage cannot be obtained easily for real systems; in [27], the voltage feed-forward method was utilized to calculate the low-pass filter bandwidth of the point of common coupling (PCC). This paper proposes a novel instantaneous frequency detecting method, in which grid current feedback control is given by an incomplete derivation with a high-pass filter. The proposed methods are shown in Figure 5. The DC bus is supported by DC sources as an ideal DC voltage. The control strategy consists of four parts, as follows.

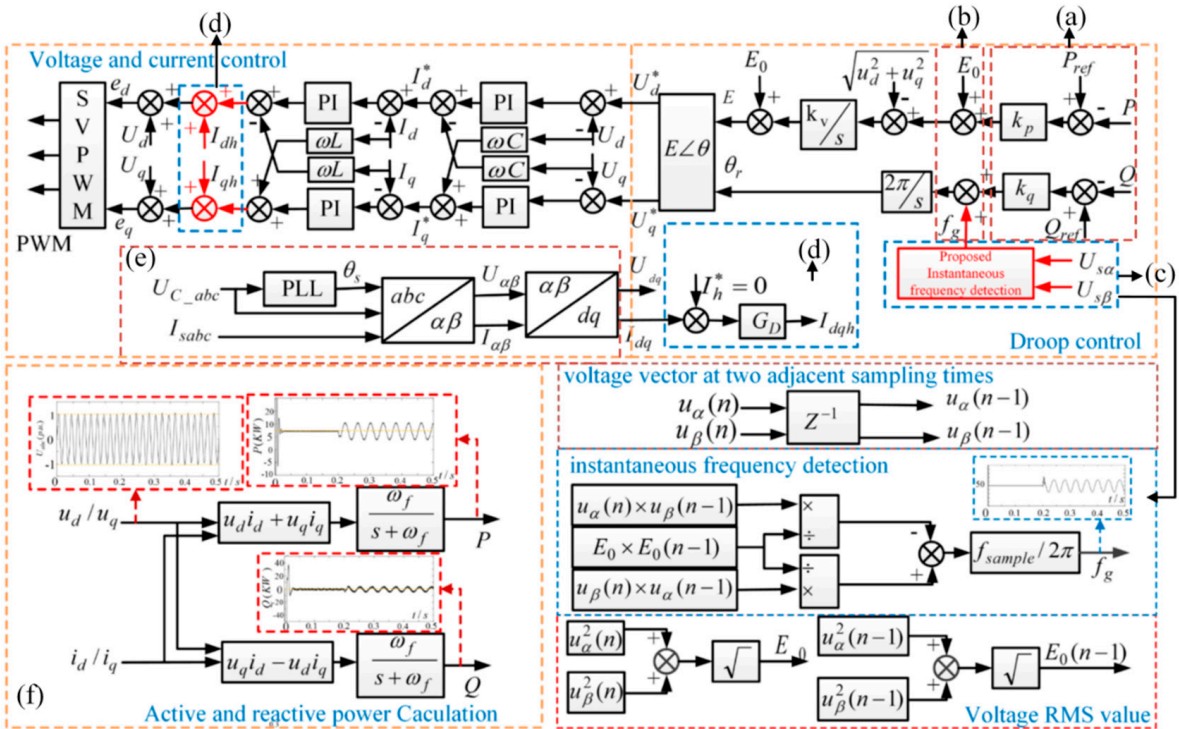

**Figure 5.** Proposed feedback control strategy. (**a**) Fundamental droop control; (**b**) feed-forward of instantaneous grid frequency and voltage magnitude; (**c**) instantaneous frequency detection method; (**d**) incomplete differential control; (**e**) coordinate transformation; (**f**) active and reactive power calculation with inter-harmonics.

(1) Coordinate transformation: The proposed control strategy is developed under the $dq$ frame, hence the fundamental parts behave as dc components and the electrical angle of the fundamental grid voltage is obtained by $u_q$ after a low-pass filter, as shown in Figure 2, to avoid the influence of distorted grid voltage. The current $I_{dq}$ and $U_{dq}$ can be obtained in the synchronous reference frame based on grid voltage $I_{abc}$ and $U_{abc}$, shown in Figure 5a.

(2) Feed-forward of grid instantaneous frequency and voltage magnitude: Power flow control in grid-connected mode may be influenced by fluctuation of grid frequency and voltage magnitude. Meanwhile, a tracking error exists in the reactive power control loop. The feed-forward of grid frequency and voltage magnitude is applied to suppress grid fluctuation impacts. Frequency is obtained by instantaneous frequency detection method, as described in Section 2.2 and shown in Figure 5b, instead of PLL.

(3) Fundamental droop control (shown in Figure 5c): According to the description in Section 2.1, the power flow control in the grid-connected mode of traditional droop control is mainly affected by two aspects—reactive power tracking static error and grid frequency and grid voltage amplitude fluctuations.

After grid voltage orientation is employed, the fundamental component of $u_{sq}$ is zero. Thus, the relationship between power and voltage and current in *dq* frame can be expressed as Equation (10), $P_s{}^*$ and $Q_s{}^*$ refer to active and reactive power.

$$\begin{cases} I_{sd}^* = P_s^*/U_{sd} \\ I_{sq}^* = Q_s^*/U_{sq} \end{cases} \tag{10}$$

When voltage is harmonically distorted, the $\alpha$-axis voltage also contains harmonic components. From Equation (10), the $\alpha$-axis reference contains the same order harmonics as the voltage. The DC component can be restrained by the PI regulator, but the harmonic components cannot be restrained due to the limited control bandwidth [28]. The main reason for grid current distortions of the inverter connected to the grid is a certain degree of the grid voltage, and the output impedance is relatively small. The strategy to suppress the harmonics of the grid current is based on increasing the harmonic impedance and counteracting the grid harmonic voltage.

If the grid voltage $u_{abc}$ contains 280 Hz inter-harmonics, as shown in Figure 5d, after power calculation, *P* and *Q* also contain 280 Hz inter-harmonic components, and they cannot be filtered by a low-pass filter.

(4)　Incomplete Differential Control

Differential control is employed to increase current control gain within a wide harmonic frequency range. However, differential control has a potential instability problem as a result of infinite gain at the infinite frequency. Thus, the differential controller is replaced by an incomplete differential element, which is expressed in Equation (11). The proposed controller is used to decrease the magnitude of the harmonic current. $H_p$ is a high-pass filter to avoid the influence of a fundamental frequency that behaves in the same way as DC components in the dq frame. Therefore, a second-order high-pass filter is preferable. $G_D$ is used to improve harmonic suppression capability of the grid-connected inverter, which is named as an incomplete derivation feedback controller. The difference between the reference of the harmonic current $I_h{}^*$ and the grid-connected output current is multiplied with $G_D$ to generate a correction term. Harmonic current $I_h{}^*$ is set to zero to suppress harmonic and inter-harmonic current. $G_D$ can be expressed as

$$G_D = H_p H_D = \frac{s^2}{s^2 + 2\xi\omega_n s + \omega_n^2} \frac{ks/\omega_f}{1 + s/\omega_f} \tag{11}$$

where $\xi$ and $\omega_n$ are the damping ratio and cutoff frequency of high-pass filter $H_p$, respectively; $H_p$ is used to separate the fundamental and harmonic frequency. In this paper, we consider harmonics as between 250–1000 Hz, $\xi$ and $\omega_n$ are 0.707 and $500\pi$ rad/s. $H_D$ is the proposed incomplete differential transfer function, the cutoff frequency $\omega_f$ should be higher than the harmonic frequency, and it is set to $10,000\pi$ rad/s; *k* is control gain.

The voltage reference in the *dq* frame comprises of the output fundamental frequency voltage $U_{PI}{}^*$, decoupling component $j\omega L I_{dq}$, and harmonic voltage $U_h{}^*$, as shown in Figure 6.

The three-phase grid-connected inverter has the same structure and no coupling term under $\alpha\beta$ frame compared with *dq* frame. The complete expression can be derived as.

$$[i_{\alpha\beta}(s)] = \frac{T(s)}{1 + T(s)} \frac{1}{H_i(s)} [i_{\alpha\beta}^*(s)] - \frac{G_{x2}(s)}{1 + T(s)} [v_{g\_\alpha\beta}(s)] \tag{12}$$

$$G_i(s) = \frac{T(s)}{1 + T(s)} \frac{1}{H_i(s)} \tag{13}$$

$$G_u(s) = \frac{G_{x2}(s)}{1 + T(s)} \tag{14}$$

According to Figure 6, the improved closed-loop transfer function of $I_{dq}$ can be derived as:

$$\begin{cases} I_{dq} = YU_{dq} + HI_{dq}^* \\ Y(s) = \frac{G_u(s)}{1+G_u(s)[G_{PI}(s)+G_D(s)]} \\ H(s) = \frac{G_u(s)G_{PI}(s)}{1+G_u(s)[G_{PI}(s)+G_D(s)]} \end{cases} \tag{15}$$

where $Y$ is transfer function of the current to the grid voltage with incomplete differential feed-forward control.

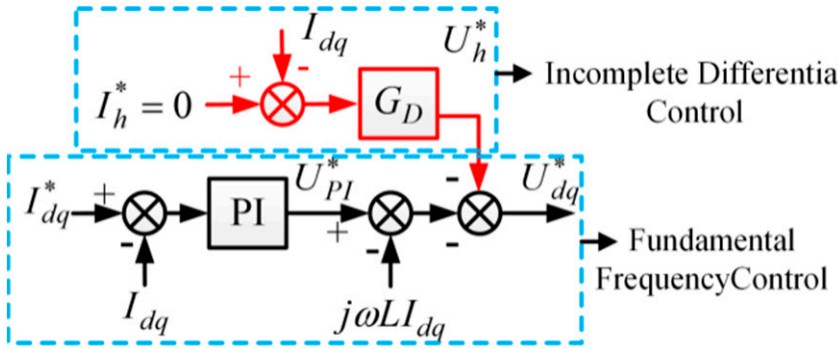

**Figure 6.** Control block diagram of $U_{dq}$.

Since $k$ in $G_D$ plays a decisive role in obtaining the harmonic suppression effect, Figure 7 shows the Bode diagram of $G_D$ under different values of $k$; the system parameters are shown in Table 1 in Section 3.

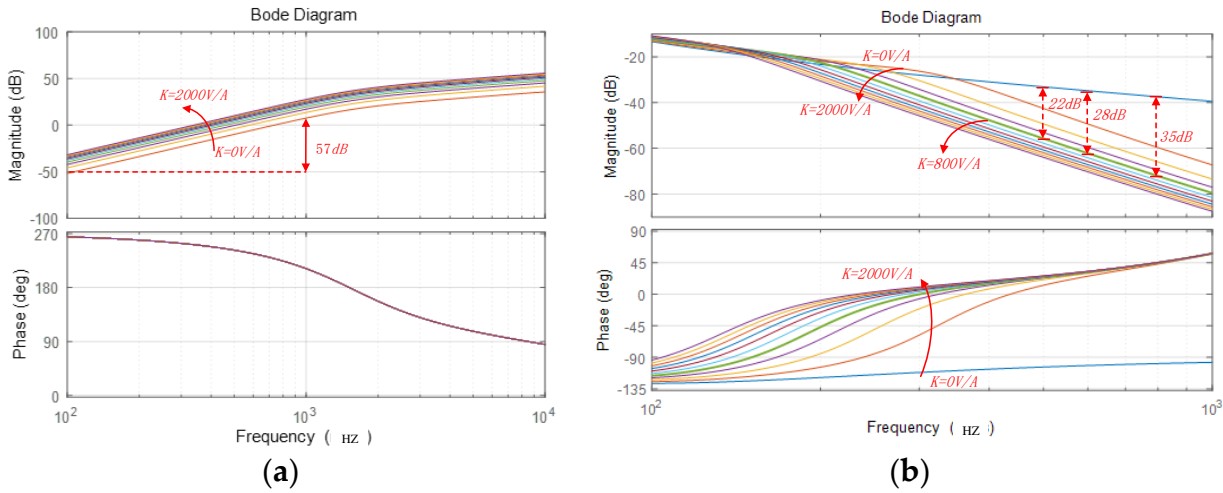

**Figure 7.** (**a**) Bode diagram of $G_D(s)$ with different control gains $k$; (**b**) Bode diagram of $Y(s)$ with different control gains $k$.

The fundamental frequency used in this paper is 50 Hz. From Figure 7, between 100–1000 Hz, increasing $k$ can magnify the amplitude of $G_D$, when $k = 100$–2000 V/A, $G_D$ is −69 dB and −50 dB at 50 Hz, which indicates that an overlarge effect may affect control performance at 50 Hz. In Figure 8, $Y(s)$ reflects the level of suppression of harmonic currents of the proposed control strategies. When $k = 0$ and $G_D(s) = 0$, it is shown that fundamental control loop has little ability for harmonic suppression, however, the proposed controller can decrease the amplitude of $Y(s)$ significantly; when $k$ is higher harmonic current suppression is improved. According to the analysis in Figure 8, $k = 800$ V/A was selected for both harmonic suppression and stability.

**Table 1.** Parameters of grid connected inverter system.

| Parameter | Values |
|---|---|
| Rated power, frequency | 10 kW, 50 Hz |
| Inductance and capacitance of LC filter | $L = 1.1$ mH, $C = 49$ μF |
| Switching frequency | $f_c = 10$ kHz |
| Controller sampling time | $T_C = 10^{-4}$ s |
| DC-link voltage | $V_{dc}^* = 600$ V |
| Grid nominal voltage (RMS) | $V_g = 400$ V |
| Transformer ratio | $n = 1:2$ |
| Grid equivalent impedance | $R = 2$ mΩ |
| Voltage inner loop coefficient | $K_P = 0.1$, $K_i = 200$ |
| Droop control coefficient | $k_P = 3 \times 10^{-3}$, $k_Q = 2 \times 10^{-4}$ |
| incomplete differential controller ($G_D$) | $k = 800$, $\xi = 0.707$, $\omega_n = 500 \times \pi$ |

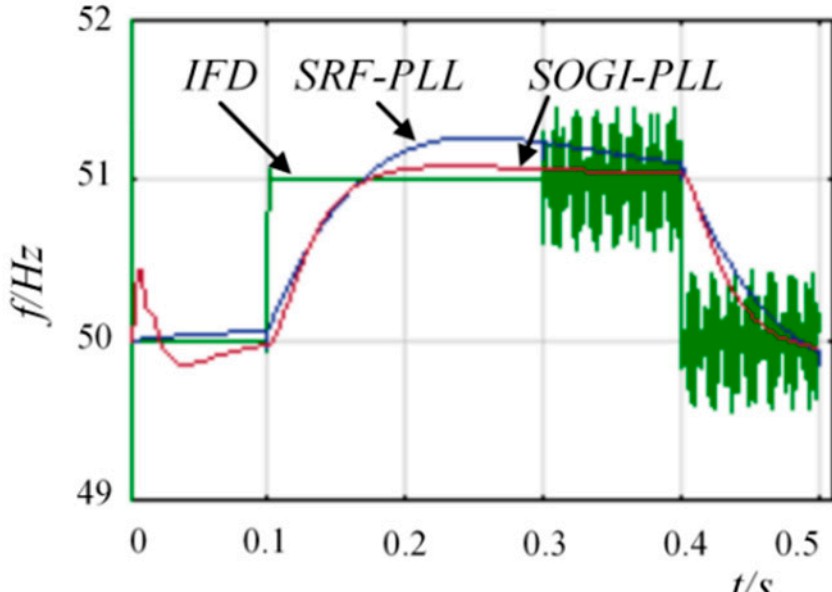

**Figure 8.** Frequency detection results.

After employing the proposed incomplete differential controller, the close-loop transfer function of $H$ shown in Equation (15) changed. For the minimum phase system, the necessary conditions for system stability included a phase margin greater than 0 and gain margin greater than 1. From Equation (15), when $k = 800$, the gain margin is 13 dB, which guarantees the stability.

## 4. Results

The performance of proposed strategies was first verified by simulations in MATLAB/Simulink, and then realized in a laboratory. A 10 kW inverter was adopted. The proposed method was compared against the conventional droop control with frequency feedforward measured by SRF-PLL without grid current feedback. According to "GB/T12668.2-2002 Speed Control Electric Drive System General Requirements Low Voltage AC Variable Frequency Electric Drive System Rated Value Regulations" in China, the definition of THD includes inter-harmonics and DC components.

### 4.1. Simulation Results

According to the structure of the grid-connected inverter system in Figure 1, the main parameters are given in Table 1. In order to simulate real system, simulations were carefully carried out in a discrete domain with a sample period of 0.1 ms. To implement the

discretization of an incomplete differential controller, a pre-warped Tustin transformation was used to convert from *s*-plane to *z*-plane.

In order to verify the proposed inter-harmonic suppression strategy for the grid-connected inverter, the simulation grid model consisted of harmonic voltage and inter-harmonic voltage with fundamental frequency fluctuation.

Case 1: Instantaneous Frequency Detection

Fluctuation of grid voltage fundamental frequency was 1 Hz and magnitude was 10 V from 0.1 s to 0.4 s, grid voltage consisted of 5.6th and 7th components of 0.0008 p.u. The voltage frequency detection results are shown in Figure 8 and compared with SRF and SOGI PLL (dual second-order generalized integrator PLL), the proposed instantaneous frequency detection method had a fast response speed and high precision.

Figures 9 and 10 show the results of the power stepping of the grid-connected inverter under harmonic and inter-harmonic conditions, enabled by the proposed control strategy. The reference of the output active power increased to 6 kW from 8 kW at 0.2 s and reference of reactive power was adjusted from 0–2 kW at 0.2 s. The reference of active power decreased to 6 kW from 8 kW. It can be seen in Figure 10 compared with Figure 9 that the proposed control strategy reactive responded accurately, it did not worsen the dynamic and steady performance of the droop control of the inverter system.

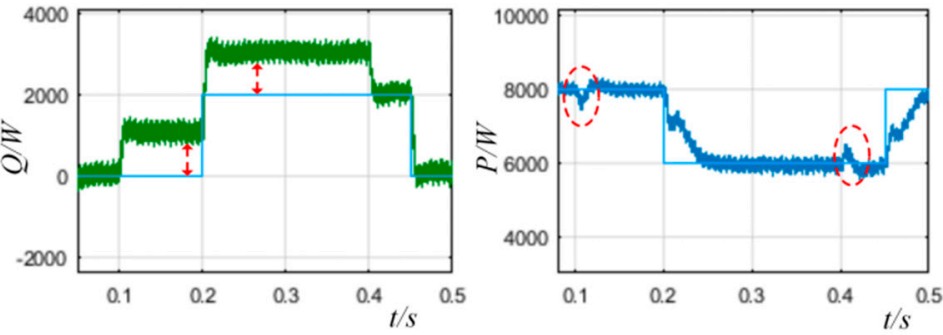

**Figure 9.** Active and reactive power under grid voltage frequency fluctuation with conventional droop control strategy.

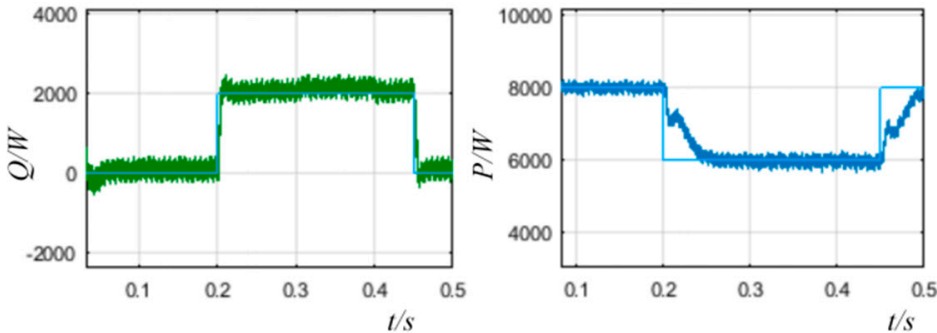

**Figure 10.** Active and reactive power under grid voltage frequency fluctuation with proposed control strategy.

Case 2: Current suppression

Initially, grid voltage fundamental frequency was set to 50 Hz and the fundamental phase voltage was 230 V. The grid voltage consisted of 250 Hz and 350 Hz, which were 0.1 p.u. and 0.1 p.u. injected at 0.2 s, and 280 Hz, 380 Hz, which were 3.5% and 3.5% injected at 0.35 s. The grid voltage, frequency, and current waveform results from phase A are shown in Figure 11a. The proposed scheme was based on the instantaneous frequency

detection described in Section 3, the traditional scheme was based on SRF-PLL and without the incomplete differential element of $k = 0$ in $G_D$.

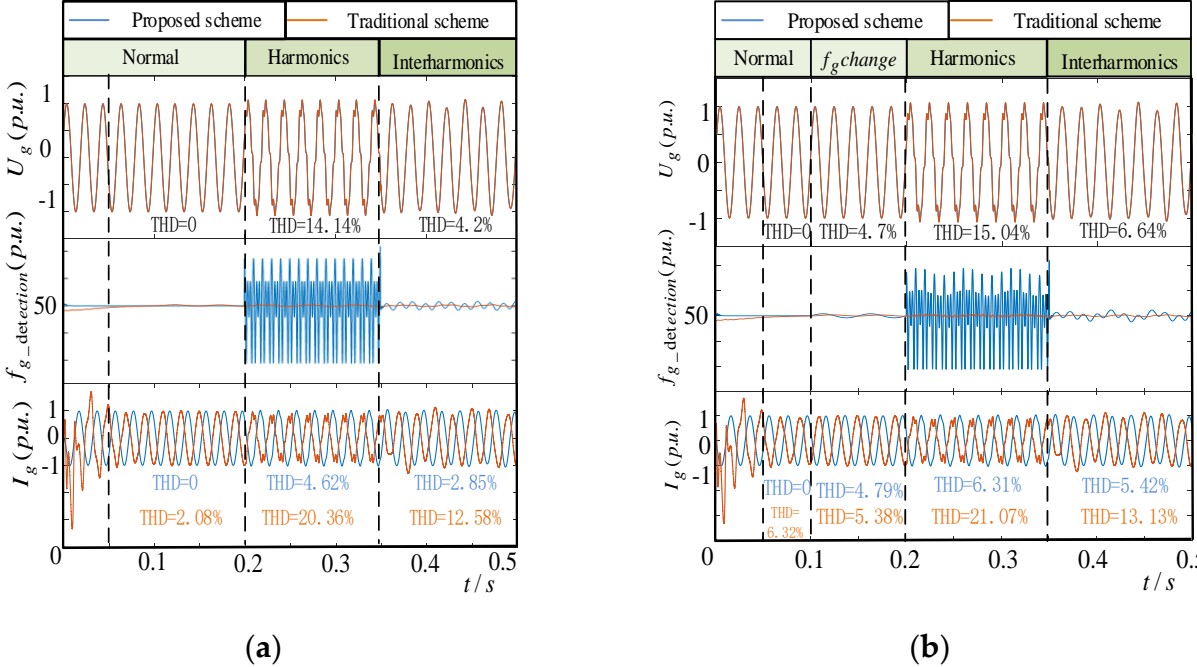

**Figure 11.** (**a**) Grid voltage waveform, output current of inverter waveform frequency, and feed-forward frequency detection results from phase A (measured under p.u.) when the grid voltage frequency was 50 Hz; (**b**) grid voltage waveform, output current of inverter waveform frequency, and feed-forward frequency detection results from phase A (measured under p.u.) when the grid voltage frequency varied from 46 Hz to 52.5 Hz.

It can be observed from Figure 11a that the measured THD of the injected grid currents was 4.62% using the proposed scheme, which was approximately 1/5 of that of the traditional scheme based on SRF-PLL 20.36% when the THD of the grid voltage was 14.14%; the measured THD of injected grid currents was 2.85%, which was approximately also 1/5 of the traditional scheme based on SRF-PLL 12.58% when the THD of the grid voltage was 4.2%. The proposed instantaneous frequency detection method could detect power supply voltage frequency changes rapidly. SRF-PLL could not detect the changes of frequency, and the oscillation time at the beginning was 0.05 s, which was larger than the IFD. Therefore, the proposed strategies suppressed the injected grid current harmonics and inter-harmonics when the fundamental frequency was stable.

The dynamic response was studied to evaluate the transient response when the fundamental frequency varied from 46 Hz to 52.5 Hz from 0.1 s to 0.5 s. The grid voltage consisted of 250Hz and 350Hz, which were 0.1 p.u. and 0.1 p.u. injected at 0.2 s, and 280 Hz and 380 Hz, which were 3.5% and 3.5% injected at 0.35 s. The grid voltage, frequency, and current waveform results from phase A are shown in Figure 11.

From Figure 11b, it can be observed that between 0.1 s to 0.2 s, the measured THD of the injected grid current was 4.79% using the proposed scheme, which was less than that of the traditional scheme based on SRF-PLL 5.38% when fundamental frequency varied and THD of the grid voltage was 4.7%. Between 0.2 s to 0.35 s, the measured THD of the injected grid current was 6.31% using the proposed scheme, which was approximately also 1/3 of that of the traditional scheme based on SRF-PLL 21.07% when THD of the grid voltage was 15.04%. Between 0.35 s to 0.5 s, the measured THD of the injected grid current was 5.42% using the proposed scheme, which was approximately also 1/3 of that of the traditional scheme based on SRF-PLL 13.13% when the THD of the grid voltage was 6.64%. The proposed strategies could detect power supply voltage frequency variety accurately. SRF-PLL could not detect the changes of frequency, and the oscillation time at

the beginning was 0.05 s, which was larger than IFD. Therefore, the proposed strategies suppressed the injected grid current harmonics and inter-harmonics when the fundamental frequency varied.

In summary, proposed strategies based on instantaneous frequency detection offer immunity against harmonics and inter-harmonics, whereas the traditional SRF-PLL do not perform accurately. In addition, the dynamic response of the proposed scheme is faster than the traditional scheme.

### 4.2. Experimental Results

The performance of proposed strategies was further verified through experimental results in laboratory. The algorithm was designed using TI-DSP28335, voltage and current were sampled by the signal acquisition and conditioning circuit and fed back to the DSP controller, and the DSP calculated the feedback amount to obtain the PWM duty cycle. Since the power of the PWM signal output by the DSP could not drive the IGBT, it was necessary to add a drive circuit to amplify the power of the PWM signal and isolate the main circuit from the control circuit to protect the control circuit. The IGBT adopts the intelligent switch module IPM. When the IGBT has an over-current fault, it will send a fault signal to the DSP error joint prevention module to prohibit the PWM output.

As shown in Figure 12, grid voltage was simulated by a programmable AC source (AN99030A Ainuo) and the proposed strategies were tested on a prototype two-level PWM inverter. A non-liner load was the source of the harmonics/inter-harmonics. The programmable AC source can change the fundamental frequency and amplitude of the grid.

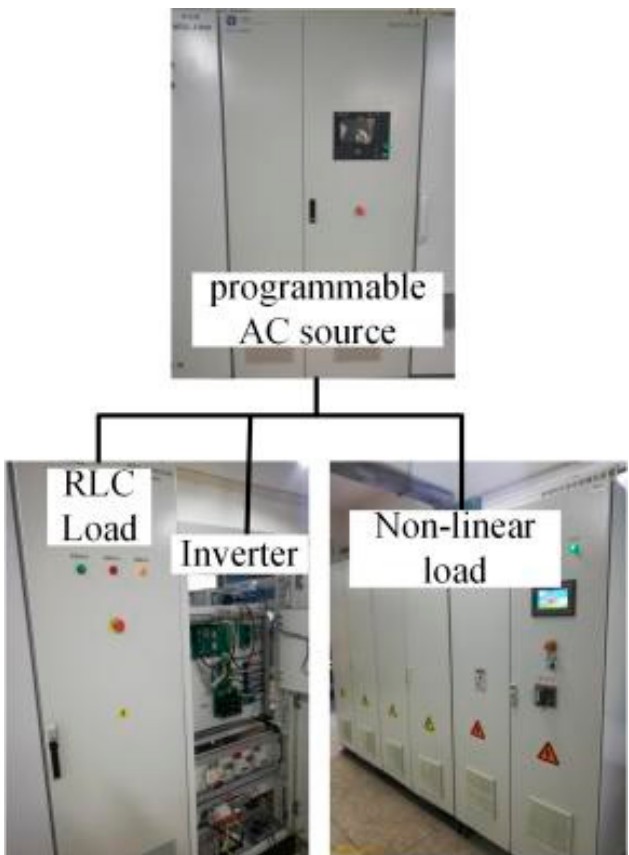

**Figure 12.** Experimental prototype developed in the laboratory.

Experimental results were measured under 3 kW active power and 2 kVar reactive power load condition. The simulated grid voltage distorted by 2% and 0.8% of the 5th and 7th harmonics and 1% of the 5.8th inter-harmonic, respectively. The fundamental

phase-to-phase voltage was 230 V, the frequency was 50 Hz, the frequency changed 0.5 Hz at 0.4 s, and the rise time was 0.1 s. We compared the output current of experimental wave forms between the traditional and proposed feed-back control methods, and the results are shown in Figure 13.

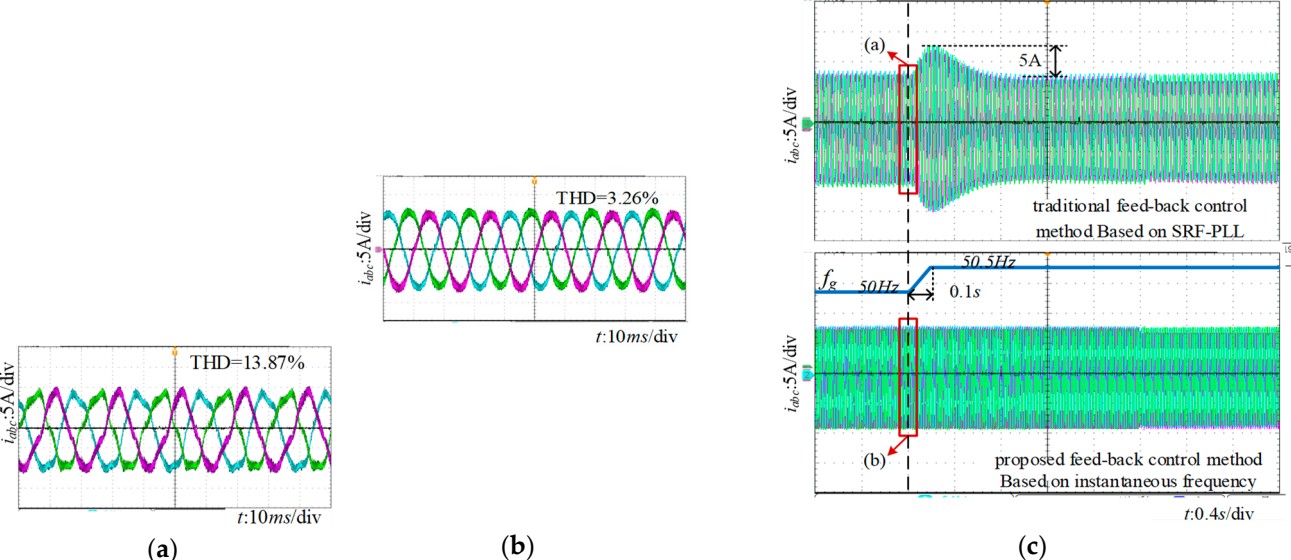

**Figure 13.** Experimental results of grid current comparison. (**a**) current responses of the traditional control method when frequency changes. (**b**) current responses of the proposed method when frequency changes. (**c**)current responses waveform.

From Figure 13c, the grid voltage fundamental frequency changed from 50 to 50.5 Hz with harmonics, and the corresponding current responses of the proposed harmonic current suppression control method was observed to be faster and more stable compared with the traditional feed-back control method. The maximum oscillation value of the current reached 5 A of the traditional feed-back control method based on SRF-PLL. The measured total harmonic distortion (THD) of the injected grid current is shown in Figure 13a,b, and the measured THD values of the injected grid currents were 13.87% and 3.26%, respectively. It can be observed that the grid voltage distortion affected the injected grid current of the micro-gird inverter and the proposed harmonic current suppression control scheme suppressed the injected grid current harmonics and inter-harmonics. Compared with the traditional feed-forward control method, the proposed strategy performed better in terms of suppression of the injected grid current harmonics.

## 5. Conclusions

The aim of this paper was to suppress the inter-harmonic current of a grid-connected droop control-based inverter using incomplete current differential feedback under grid voltage conditions with inter-harmonics and frequency fluctuations. Conclusion can be highlighted as follows.

1.  The proposed instantaneous frequency detection method has a faster response and better precision than SRF- and SOGI-PLL systems under grid voltage conditions with fundamental frequency fluctuations and inter-harmonics.
2.  The suppression strategy based on incomplete differential feedback can work on a harmonic range, so that the harmonic frequency detection can be avoided and inter-harmonic current can also be suppressed.
3.  The incomplete differential feedback and instantaneous frequency feed-forward methods were applied in a droop control; control parameters can be designed by considering the harmonic current suppression frequency range. This method is flexible and convenient to apply to future inverter custom modifications.

**Author Contributions:** Conceptualization, W.X. and B.W.; methodology, W.X.; software, W.X. and D.L.; validation, W.X., J.L., and D.L.; formal analysis, W.X.; investigation, W.X.; resources, B.W.; data curation, W.X.; writing—original draft preparation, W.X.; writing—review and editing, W.X.; visualization, J.L.; supervision, B.W.; project administration, B.W.; funding acquisition, B.W. All authors have read and agreed to the published version of the manuscript.

**Funding:** The work described in this paper was supported in part by the National Natural Science Foundation of China (No.51877161 to Bin Wang) and Teaching Research Project (No.2020X066 to Wanwan Xu). The authors declare that they have no competing financial interests.

**Data Availability Statement:** All data included in this study are available upon request by contact with the corresponding author.

**Conflicts of Interest:** We declare that we have no financial and personal relationships with other people or organizations that can inappropriately influence our work.

## Appendix A

**Table A1.** Definition of symbols in Figure 1.

| Definition | Symbols |
| --- | --- |
| Voltage of DC side | $U_{dc}$ |
| Reference voltage | $V_{ref}$ |
| Grid voltage | $u_g$ ($V_g$ as RMS) |
| Output voltage of inverter | $v_{abc}$ ($V_{abc}$ as RMS) |
| Output current of inverter | $i_{2abc}$ |
| Conductance and capacity of filter | $L_1$, $L_2$, $C_f$ |
| sampling coefficient | $H_1$, $H_2$ |
| Instantaneous active power, reactive power | $P$, $Q$ |
| Reference active power, reactive power | $P_{ref}$, $Q_{ref}$ |
| Grid equivalent impedance | $L_g = R_g + jX_g$ |

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
