# Peer review of "An Improved Droop Control Strategy for Grid-Connected Inverter Applied in Grid Voltage Inter-Harmonics and Fundamental Frequency Fluctuation"

_electronics, doi:10.3390/electronics10151827_

Round 1

Reviewer 1 Report

The manuscript proposes a solution to the very important problem of controlling inverters in the presence of inter-harmonics as well as frequency fluctuations. The instantaneous frequency detection method proposed in the manuscript is indeed very promising and should be of value to the community. The simulation and experimental results seem to validate the proposed methods as well.

However, the manuscript has several typographical errors and does not explain the feed-forward method to suppress harmonics in output currents clearly. I believe there are several unanswered questions, especially with respect to the feedforward method. The section titled 'Inter-harmonic suppression control strategy' (section 2? 3?) should be significantly modified such that the proposed method is clearer. In the attached document, I have highlighted several lines with my questions as well as possible errors and unclear sentences. Some of the major questions which I have are:

  1. If the dq-frame is used for the proposed method, how is the 'phase-angle' (the dynamic angle between the d-axis and the a-axis) used for conversion from ab to dq frame computed, especially under unknown/interharmonic frequency?
  2. I am unaware of the technique of 'differential control' which is being discussed in the paper. Need to have a reference to clarify what this technique means, since I do not believe this is a widely known method. 
  3. How is THD defined when we have inter-harmonics, since conventional definition of THD involves only harmonic multiples? It would be useful to discuss how this is numerically computed for the case of inter-harmonics.

I believe that the manuscript needs significant linguistic improvements and better description of the feedforward method in order to be of valuable contribution towards the problem it discusses. Hence, I recommend that only after major revisions may the manuscript be accepted for publication.

Reviewer 2 Report

This paper presented presents a current suppression method based on droop control strategy under distorted grid voltage with inter-harmonics and fundamental frequency fluctuation. The overall quality of the paper is acceptable with a few remarks.

Comments and remarks:

  • Most of the figures are unreadable, please, correct them with a focus on the higher quality.
  • Please, check up on the list of symbols, the meaning of a lot of them is unclear, f.e. L is inductance or impedance or conductance?
  • The presented droop control strategy needs to be described in more detail. In particular, the aim should be to distinguish it from the control used

Author Response

Dear Editors and Reviewers:

Thank you for your letter and for the reviewers’ comments concerning our manuscript entitled “An Improved Droop Control Strategy for Grid-connected Inverter Applied in Grid Voltage Inter-harmonics and Fundamental Frequency Fluctuation” (ID: electronics-1293307). Those comments are all valuable and very helpful for revising and improving our paper, as well as the important guiding significance to our researches. We have studied comments carefully and have made major revisions which we hope this revision can make our paper more acceptable. The main corrections in the paper and the responds to the reviewer’s comments are as flowing:

(Revised portion are marked in red in the paper.)

Main corrections:

In Introduction part:

  • we have added newliterature ‘Two-Dimensional Impedance-Shaping Control With Enhanced Harmonic Power Sharing for Inverter-Based Microgrids’ given by reviewer. The paper analyzed harmonic power sharing caused by mismatched grid impedance for island micro-grid, then developed a two-dimensional impedance shaping control. Considering circulation current or instability problem in multiple inverters, we only considered the harmonic suppression of a single interlinking inverter.
  • We have modified the comparisonwith existing works from line 73 to 86.

In Section 1

  • formulas (5) and (6) have been modified; Symbols’descriptions of formulas and figure 1 have been added.
  • Based on reviewer #1’s comments,we have revised the correct or incomplete sentences in this revision.

In Section 2

  • The section title have been modified to ‘Control Strategy based on current incomplete differential feedback’.
  • Thecontrol strategy consists of four parts were described in more detail. They are ‘Coordinate Transformation, frequency feed-forward, fundamental droop control, Incomplete differential control’.
  • We have modified figure 5and figure 6 for more readable.

In Result part 2

  • we have added a description of the definition of THD, that is "GB/T12668.2-2002 Speed Control Electric Drive System General Requirements Low Voltage AC Variable Frequency Electric Drive System Rated Value Regulations" in China.
  • Re-described some experimental results and comparison.
  • we have checked up meaning of symbols in this revision.

In addition, this version have reviewed by my friends Bismark, international student from UK.

Reviewer #2:

This paper presented presents a current suppression method based on droop control strategy under distorted grid voltage with inter-harmonics and fundamental frequency fluctuation. The overall quality of the paper is acceptable with a few remarks.

1) Most of the figures are unreadable, please, correct them with a focus on the higher quality.

Author’s response:

Thanks for reviewer’s suggestions. Figures and charts play an important role in papers. We have modified figure 5 and figure 6 for more readable.

 2) Please, check up on the list of symbols, the meaning of a lot of them is unclear, f.e. L is inductance or impedance or conductance?

Author’s response:

We are very sorry for our negligence of some symbols in this paper, we have checked up in this revision. L and C in table 1 are inductance and capacitance of LC filter respectively.

 3) The presented droop control strategy needs to be described in more detail. In particular, the aim should be to distinguish it from the control used

Author’s response:

Many thanks for this comment. Here we present a harmonic current suppression strategy based on droop control which has been validated within a harmonic frequency range instead of at frequency points. We have rewritten the Section 2 and Results part for more discussed. The control strategy consists of four parts were described in more detail. They are ‘Coordinate Transformation, frequency feed-forward, fundamental droop control, Incomplete differential control’.

Thank you very much for your all comments.

Best regards!

Reviewer 3 Report

This paper proposes a current suppression method based on droop control strategy under distorted grid voltage with inter-harmonics and fundamental frequency fluctuation. The comments of this paper are as follows:
1. The literature review of harmonics compensation and sharing is not sufficient. A more comprehensive literature review and categories of each method are suggested. Some recent works can be referred: doi: 10.1109/TPEL.2019.2898670. 
2. The contributions should be justified by comparing them with existing works. 
3. There is two separate design to overcome frequency fluctuation and inter harmonics suppression. The overall system performance and stability should be further discussed. 
4. If multiple inverters adopted the proposed method simultaneously, is there any circulation current or instability problem?
5. The authors should compare the results with some recently published works. 

Author Response

Dear Editors and Reviewers:

Thank you for your letter and for the reviewers’ comments concerning our manuscript entitled “An Improved Droop Control Strategy for Grid-connected Inverter Applied in Grid Voltage Inter-harmonics and Fundamental Frequency Fluctuation” (ID: electronics-1293307). Those comments are all valuable and very helpful for revising and improving our paper, as well as the important guiding significance to our researches. We have studied comments carefully and have made major revisions which we hope this revision can make our paper more acceptable. The main corrections in the paper and the responds to the reviewer’s comments are as flowing:

(Revised portion are marked in red in the paper.)

Main corrections:

In Introduction part:

  • we have added newliterature‘Two-Dimensional Impedance-Shaping Control With Enhanced Harmonic Power Sharing for Inverter-Based Microgrids’ given by reviewer. The paper analyzed harmonic power sharing caused by mismatched grid impedance for island micro-grid, then developed a two-dimensional impedance shaping control. Considering circulation current or instability problem in multiple inverters, we only considered the harmonic suppression of a single interlinking inverter.

  • We have modified the comparisonwith existing works from line 73 to 86.

In Section 1

  • formulas (5) and (6) have been modified; Symbols’descriptions of formulas and figure 1 have been added

  • Based on reviewer #1’s comments,we have revised the correct or incomplete sentences in this revision.

In Section 2

  • The section title have been modified to ‘Control Strategy based on current incomplete differential feedback’.

  • Thecontrol strategy consists of four parts were described in more detail. They are ‘Coordinate Transformation, frequency feed-forward, fundamental droop control, Incomplete differential control’.

  • We have modified figure 5and figure 6 for more readable.

In Result part 2

  • we have added a description of the definition of THD, that is "GB/T12668.2-2002 Speed Control Electric Drive System General Requirements Low Voltage AC Variable Frequency Electric Drive System Rated Value Regulations" in China.

  • Re-described some experimental results and comparison.

  • we have checked up meaning of symbols in this revision.

In addition, this version have reviewed by my friends Bismark, international student from UK.

Reviewer #3:

  1. The literature review of harmonics compensation and sharing is not sufficient. A more comprehensive literaturereview and categories of each method are suggested. Some recent works can be referred: doi: 10.1109/TPEL.2019.2898670.

Author’s response:

Many thanks for reviewer’s carefully revising our paper and collecting literature for us. We have studied ‘Two-Dimensional Impedance-Shaping Control With Enhanced Harmonic Power Sharing for Inverter-Based Microgrids’. This paper focused on circulating harmonic currents for parallel voltage source inverters. Our paper concerned on harmonic current suppression under grid voltage inter-harmonics and frequency fluctuation, the grid impedance and multiple inverters are not researched in this paper. However, these are important research points for our future work, and we will continue our research on hybrid DC/AC interlinking inverter for considering grid impedance and multiple inverters.

Meanwhile, we also have added this literature in our References.

  1. The contributions should be justified by comparingthem with existing works.

Author’s response:

Many thanks for this comment. The existing works of harmonic current suppression were based on repetitive control or resonant regulator works on integer harmonic voltage effectively, but this is not a valid approach for inter-harmonic voltage. Meanwhile, these method considers the fundamental frequency as fixed. We have modified the comparison in Introduction part.  

  1. There is two separate design to overcome frequency fluctuation and inter harmonics suppression. The overall system performance and stability should be further discussed.

Author’s response:

we appreciate the Reviewer’s carefully reading. Here we present a harmonic current suppression strategy which has been validated within a harmonic frequency range instead of at frequency points. The value of frequency feed-forward is calculated by the rotating angular velocity of the voltage vector instead of phase-locked loop (PLL) with fundamental frequency variations. Two methods are concentrated in one control system, we have rewritten the Section 1.2 and Results part for more discussed.

  1. If multiple inverters adopted the proposed method simultaneously, is there any circulation current or instability problem?

Author’s response:

As the power level of the distributed generation system increases, the interface is reversed. Parallel operation of inverters has become an inevitable choice.The use of direct current control will cause parallel circulating current problems for grid-connected inverters. In island operation mode, each inverter is divided according to its own rated capacity, mismatch of line impedance between inverters of different rated capacity will cause uneven load distribution in the power grid, causing circulation phenomenon.

In the actual project, virtual impedance to suppress the circulating current has been widely used. Our proposed method is based on droop control, which is different from direct current control, circulation current or instability problem will be happened when considering line impedance. Even though we did not discussed in this article, but will be our follow-up research content.

  1. The authors should compare the resultswith some recently published works.

Many thanks for this comment. We have modified the comparison with recently published works in Introduction part. Due to the modification time limitation, we cannot reproduce the experimental results of the methods in the existing literature. However, the advantages of the proposed method in harmonic compensation is suitable for grid voltage fundamental frequency change and considering inter-harmonics. The existing harmonic current suppression method for the inverter based on the resonant regulator or repetitive control only effectively works on the integer harmonic voltage with the fixed harmonic frequency, which will be invalid on the inter-harmonic voltage. The inter-harmonic frequency is usually unfixed. If the resonant regulator is employed to suppress the inter-harmonic current, the frequency of the harmonic current should be accurately detected in the real time. Discrete Fourier transform and fast Fourier transform are commonly used to detect the harmonics, while the problems of the spectral leakage phenomenon and the picket-fence effect At the same time, fundamental frequency fluctuation has not been paying attention.

Thank you very much for all your comments!

Best regards!

Round 2

Reviewer 1 Report

I believe that the authors have done a great job revising the manuscript. The explanations of the novel ideas and methods in the paper are now much more clear and accessible.

In my opinion, this manuscript tackles a very interesting and relevant problem, and offers new insights to the solution of the problem of inter-harmonic management. I recommend that the manuscript be accepted for publication.

Reviewer 3 Report

This revision works for me. I do not have further comments.

This manuscript is a resubmission of an earlier submission. The following is a list of the peer review reports and author responses from that submission.

Round 1

Reviewer 1 Report

  • I strongly recommend this valuable work for publication.
  • for grid connected inverters, several controllers have been proposed. one of the most important method is model predictive controller. the authors should study this method in introduction section using below references:
  • A New Seven-Level Grid-Connected Converter Using Model Predictive Controller
  • New Grid-Connected Multilevel Boost Converter Topology with Inherent Capacitors Voltage Balancing Using Model Predictive Controller.

Author Response

Dear Editors and Reviewers:

Thank you for your letter and for the reviewers’ comments concerning our manuscript entitled “An Improved Droop Control Strategy for Grid-connected Inverter Power Stability and Harmonic Suppression Under Grid Fluctuation and Inter-harmonics” (ID: electronics-1189426). Those comments are all valuable and very helpful for revising and improving our paper, as well as the important guiding significance to our researches. We have studied comments carefully and have made major revisions which we hope this revision can make our paper more acceptable. The main corrections in the paper and the responds to the reviewer’s comments are as flowing:

(Revised portion are marked in red in the paper.)

Main corrections:

Format and English writing

(1) The format of manuscript has been turned into standard document template from website of electronics

(2) The letters and numbers in Figures are increased size for more readable, vertical layout of Fig 1.a) and below Fig 1.b)

(3) We updated the manuscript by refining some sentences for the whole paper, especially for long sentences.

(4) Grammatical errors have been checked, such as verbs missing.

(5) We deleted some redundant abbreviations that are used only few times in this paper, such as “MG”, “PR”

Content

(1) We modified the title to “An Improved Droop Control Strategy for Grid-connected Inverter Applied in Grid Fluctuation and Inter-harmonics” for more clear focus on main subjects.

(2) We updated the manuscript by adding the comparison model of Predictive Controller in Introduction part.

(3) We rewritten the Abstract part for more readable and focused on the main subjects of the title.

(4) We re-sorted the results of existing research in Introduction part, separated their original ideas, and cleared our academic viewpoints.

(5) Section 2.2 (that is Section 1.2 in the new version) has been recomposed.

(6) Some important references have been added for Fig.2 and related contents.

(7) we updated the references for some missing years of publications in [2], [3], [6], [7], [8]

(8) We added a clear list of symbols in appendix and deleted some symbols that are not used in text.

Response 1#

Thanks for your suggestions. We are very sorry for our negligence of model predictive controller (MPC) in this paper, which is also one of our research topics in our team. Different from these two references, we focused on an adaptive model predictive controller for voltage sensor less grid-connected converter system. According to the existing literature and our research results, when properly designed for and tailored to a given case study, MPC can achieve favorable system performance. This, however, comes at the expense of pronounced computational complexity— especially when implemented naively – implying that powerful control platforms and efficient real-time solvers are required in many cases.

as Reviewer suggested, we have updated the manuscript by adding the comparison model of Predictive Controller in Introduction part line 45-48, maybe in the future, we will try to solve the problems in this article with MPC.

Special thanks to you for your good comments.

Best regards.

Reviewer 2 Report

This paper deals with a droop control in grid-connected inverter system. But, the manuscript is not well organized, and it is very hard to understand the contents because of poor English. I think that the extensive editing of English is necessarily required. In overall, the manuscript should be carefully re-organized. As a simple example, the first equation starts with the number of (4). Also, in conclusion, there is no mention about droop control.

Author Response

Dear Editors and Reviewers:

Thank you for your letter and for the reviewers’ comments concerning our manuscript entitled “An Improved Droop Control Strategy for Grid-connected Inverter Power Stability and Harmonic Suppression Under Grid Fluctuation and Inter-harmonics” (ID: electronics-1189426). Those comments are all valuable and very helpful for revising and improving our paper, as well as the important guiding significance to our researches. We have studied comments carefully and have made major revisions which we hope this revision can make our paper more acceptable. The main corrections in the paper and the responds to the reviewer’s comments are as flowing:

(Revised portion are marked in red in the paper.)

Response 2#: Thanks for your suggestions. As Reviewer suggested that extensive editing of English has been organized carefully, the main corrections are as follows. In Conclusion part, we have Re-concluded the proposed improved droop control.

Main corrections:

Format and English writing

(1) The format of manuscript has been turned into standard document template from website of electronics

(2) The letters and numbers in Figures are increased size for more readable, vertical layout of Fig 1.a) and below Fig 1.b)

(3) We updated the manuscript by refining some sentences for the whole paper, especially for long sentences.

(4) Grammatical errors have been checked, such as verbs missing.

(5) We deleted some redundant abbreviations that are used only few times in this paper, such as “MG”, “PR”

Content

(1) We modified the title to “An Improved Droop Control Strategy for Grid-connected Inverter Applied in Grid Fluctuation and Inter-harmonics” for more clear focus on main subjects.

(2) We updated the manuscript by adding the comparison model of Predictive Controller in Introduction part.

(3) We rewritten the Abstract part for more readable and focused on the main subjects of the title.

(4) We re-sorted the results of existing research in Introduction part, separated their original ideas, and cleared our academic viewpoints.

(5) Section 2.2 (that is Section 1.2 in the new version) has been recomposed.

(6) Some important references have been added for Fig.2 and related contents.

(7) we updated the references for some missing years of publications in [2], [3], [6], [7], [8]

(8) We added a clear list of symbols in appendix and deleted some symbols that are not used in text.

Special thanks to you for your good comments.

Best regards.

Reviewer 3 Report

The paper presents simulation results and experimental tests and measurements.

The title is redundant and confusing. Contains too much information.   “An Improved Droop Control Strategy for Grid-connected In-verter Power Stability and Harmonic Suppression Under Grid Fluctuation and Inter-harmonics”. ?. Consider a more compact formulation, with focus on the main subject.

The Abstract. The sentences are too long. Consider fragmenting in short and concise sentences.

In Introduction:

Avoid redundant abbreviations such as “MG”, “PR” etc, that are used only few times.

Some verbs are missing.

Regarding the sentence: “Interventionary studies involving animals or humans, and other studies that re-quire ethical approval, must list the authority that provided approval and the corre-sponding ethical approval code.”: which is the relation to the “Improved Droop Control Strategy for Grid-connected In-verter”?

In Section 2. Modeling of grid-connected inverters.

The letters and numbers in Fig 1 are too small and not readable. Consider vertical layout of Fig 1.a) and below Fig 1.b) and also increase the size of both figures a) and b).

Which is the meaning of:

 “SVG is used to reactive power compensation and voltage balancing renewable energy inte-gration, however, SVGs have complex interactions among inverters, inverters, and grid under weak grid [14].”?

“When micro-grid is in grid-connected mode.” ?

“When grid frequency increases, decreasing delivered active power; when grid voltage increases, decreasing injected reactive power.” ?

In Fig. 1 there is no “ug”, “VNN” etc. Add a clear list of symbols.

“M0 is the closed-loop transfer function matrix of the 28 system” Mo is not defined.

“GPU(s) is transfer function of grid voltage amplitude and active power” is not defined.

“ GQU(s) is transfer function of grid voltage amplitude and reactive power; GPf(s) is transfer function of grid voltage frequency and active power; GQF(s) is transfer function of grid voltage fre-quency and reactive power.” are not defined.

Fig 2 and the lines 55-59 needs the exact reference.

Fig. 5 is not readable. See my comments to Fig 1.

The text in Section 2.2. is not coherent. The sentences are copied-pasted from references, without homogenization, they are of different styles, and syntax.

In References: Missing years of publications in [2], [3], [6], [7], [8], etc. The year of publication is important. Consider to retrieve complete references.

Main objection:

The authors must clearly separate their original ideas and research carried out, from the ideas and materials retrieved from references. The originality of the paper is not clear.

Reviewer 4 Report

Congratulations. Good work!

Round 2

Reviewer 3 Report

Following up with this reviewers’ comments, the manuscript is very much improved.

The document “cover letter - responses to reviewers” contains explanations about authors’ improvements, and how they collaborated to solve the weak parts of the initial manuscript.

However, the manuscripts needs one more careful spelling verification.

Is this reviewers’ feeling that the authors like more the revised manuscript that the first submitted version, and will recommend it for publication in Electronics Journal.

Author Response

Dear Editors and Reviewers:

Thank you for your letter and for the reviewers’ comments concerning our manuscript entitled “An Improved Droop Control Strategy for Grid-connected Inverter Applied in Grid Fluctuation and Inter-harmonics” (ID: electronics-1189426). Those comments are all valuable and very helpful for revising and improving our paper. We have studied comments carefully. The main corrections in the paper are English spelling mistakes, long sentences simplify, grammatical tense errors, et al. The co-author Da Li’s email replaced by l1d2moan@126. com. Revised portion are marked in blue in the paper and the responds to the reviewer’s comments are as following.

Reviewer #3’s comment:

  • However, the manuscripts needs one more careful spelling verification. Is this reviewers’ feeling that the authors like more the revised manuscript that the first submitted version, and will recommend it for publication in Electronics Journal.

Author’s response:

Thanks for your suggestions. We are very sorry for our spelling mistakes in the second revised manuscript. After carefully checked, we have modified grammatical tense errors, such as “has” to “had”, and simplified long sentences. We thank the reviewers for the time and effort that they have put into reviewing the previous version of the manuscript. Their suggestions have enabled us to improve our work. We uploaded the file of the revised manuscript. Accordingly, we have uploaded a copy of the second revised manuscript with all the changes highlighted by bule color in MS Word. We hope that this revised manuscript is accepted for publication in Electronics Journal.

Special thanks to you for your good comments.

Best regards.

2021.5.19